# Extreme fast charging of commercial Li-ion batteries via combined thermal switching and self-heating approaches

Yuqiang Zeng[1,4], Buyi Zhang [1,2,4], Yanbao Fu[1], Fengyu Shen[1], Qiye Zheng[1,2,3], Divya Chalise [1,2], Ruijiao Miao[1,2], Sumanjeet Kaur [1], Sean D. Lubner[1], Michael C. Tucker [1], Vincent Battaglia[1], Chris Dames[1,2] & Ravi S. Prasher [1,2] ✉

The mass adoption of electric vehicles is hindered by the inadequate extreme fast charging (XFC) performance (i.e., less than 15 min charging time to reach 80% state of charge) of commercial high-specific-energy (i.e., >200 Wh/kg) lithium-ion batteries (LIBs). Here, to enable the XFC of commercial LIBs, we propose the regulation of the battery's self-generated heat via active thermal switching. We demonstrate that retaining the heat during XFC with the switch OFF boosts the cell's kinetics while dissipating the heat after XFC with the switch ON reduces detrimental reactions in the battery. Without modifying cell materials or structures, the proposed XFC approach enables reliable battery operation by applying <15 min of charge and 1 h of discharge. These results are almost identical regarding operativity for the same battery type tested applying a 1 h of charge and 1 h of discharge, thus, meeting the XFC targets set by the United States Department of Energy. Finally, we also demonstrate the feasibility of integrating the XFC approach in a commercial battery thermal management system.

The long charge time (>30 min) of electric vehicles (EVs) compared with the refueling time of gasoline vehicles has been a major barrier to the mass adoption of EVs[1-4]. Currently, the charge time to 80% state of charge (SOC) in EVs such as Tesla models with fast charging capabilities is >30 min[5]. For a recharging experience comparable to that of gasoline vehicles, called extreme fast charging (XFC) of EVs, the United States Department of Energy (US DOE) has set a goal of <15 min charge time to 80% SOC, >180 Wh/kg discharge specific energy, and <20% capacity loss in 500 XFC cycles[6,7].

It is acknowledged that long XFC cycle life cannot be achieved in existing commercial high-energy-density lithium-ion batteries (LIBs) with graphite (C) negative electrodes and transition-metal oxide positive electrodes such as lithium cobalt oxide (LCO)[2]. Reducing the charge time to 15 min requires a charge rate of 6C for the constant-current stage of Constant Current Constant Voltage (CCCV) Charging,

which can trigger lithium plating on graphite negative electrodes and cause dramatic capacity fade in LIBs. Eliminating or mitigating lithium plating[8-10], which requires faster ion transport and kinetics in LIBs, is one of the greatest research and development (R&D) challenges remaining to enable XFC. Broadly, R&D efforts to develop XFC LIBs can be classified into four categories: the development of new electrolytes[11,12], electrode materials[13-17], charge protocols[18,19], or heating strategies (i.e., improving the kinetics by increasing the temperature before XFC)[20-24]. Among these approaches, heating strategies have shown promising results for existing high-energy-density LIBs and thus have the potential to enable XFC of EVs in the near term.

Treating the battery as a lumped thermal system (see Supplementary Note 1 for details on the validity of lumped model), the transient battery temperature can be written as $T_B(t) = (Q - mC_p \frac{\partial T_B}{\partial t})/hA + T_c$, where $Q$, $A$, $T_B$, $m$, and $C_p$ are the

[1]Energy Storage and Distributed Resources Division, Lawrence Berkeley National Laboratory, Berkeley, CA 94720, USA. [2]Department of Mechanical Engineering, University of California, Berkeley, CA 94720, USA. [3]Mechanical and Aerospace Engineering Department, The Hong Kong University of Science and Technology, Hong Kong, China. [4]These authors contributed equally: Yuqiang Zeng, Buyi Zhang. ✉e-mail: rsprasher@lbl.gov

transient heat generation, surface area, temperature, mass, and heat capacity of the battery, respectively. $T_C$ is the coolant temperature and $h$ denotes the tunable thermal conductance per unit area between the battery and coolant, as illustrated in Fig. 1a. Thus, elevating the battery temperature relies on the increased $Q$ and $T_C$ and/or the reduced $hA$ (see Supplementary Note 1 for the 3D transient electrochemical-thermal model). Two heating strategies have been proposed/enacted for fast charging. (1) The first approach is system-level $T_B$ control using battery thermal management systems (BTMSs)[22,24] by adjusting $h$ and $T_C$ using coolant modulation ("CM" in Fig. 1a), e.g., increasing or reducing $h$ by starting or stopping the coolant flow and/or changing $T_C$ by heating or cooling the coolant. $T_B$ is raised by reducing $h$ and/or increasing $T_C$ during fast charging and reduced during rest and discharge by increasing $h$ and/or reducing $T_C$. In fact, coolant-controlled charge protocols are being adopted by EV companies[22]; however, the maximum charge rate is only 2C (where 1C represents 1 h of testing to charge or discharge LIBs fully) as opposed to the possibility of 4C–6C

suggested by electrochemical-thermal (ECT) simulations[24]. The difference in rate capability between the simulation and the real world stems from the low gravimetric (0.55–0.65) and volumetric (<0.4) cell-to-pack (CTP) ratio in practical battery packs[23]. Based on our validated ECT model of battery packs with such low CTP ratios (Supplementary Fig. 1 and Supplementary Note 1), a large portion of battery heat (~40%) is dissipated to the pack due to its high thermal mass even if coolant flow is completely stopped, thereby limiting the temperature rise during charging. This results in negative electrode potentials below 0 V vs. Li/Li⁺, indicating the high likelihood of lithium plating. (2) The second strategy is cell-level $T_B$ control with increased $Q$ using embedded nickel foil heaters and reduced $h$ by thermally insulating the cell ("Insulation" in Fig. 1a)[21]. This method enables higher C rates as the increase of $T_B$ is observably higher than that for the system-level strategy. However, there are two major challenges with this method. Because the battery is always thermally insulated[21], $T_B$ is high even during discharge and rest, and the high average $T_B$ during operation

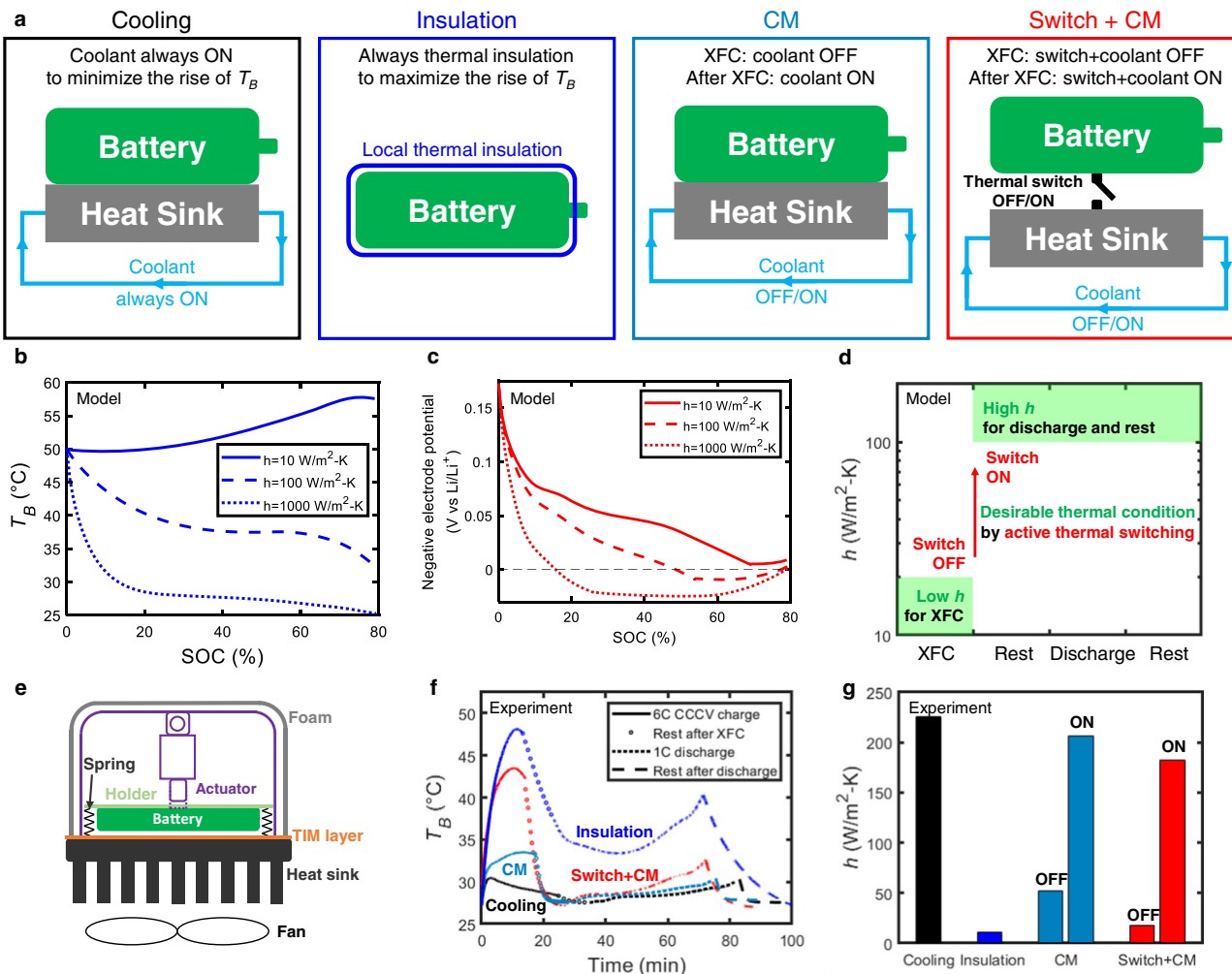

**Fig. 1 | Active thermal switching for XFC. a** Schematic of different thermal protocols for XFC. Note CM refers to coolant modulation (see text for details). The concept proposed in this paper combines CM with an active thermal switch. Prediction of battery **b** temperature and **c** negative electrode potential during 6C (−13 mA/cm²) charging of 10-Ah C‖LCO pouch cells with an initial $T_B$ of 50 °C and different heat transfer coefficients. Insufficient thermal insulation can result in a temperature decrease and negative electrode potentials below 0 V vs. Li/Li⁺ during XFC. **d** Thermally modulated charge protocol (TMCP) with thermal switching ratio ≥10 for XFC, designed with our ECT model. Note that in the OFF state of the thermal switch, the coolant flow was also OFF, whereas in the ON state, the coolant flow was also ON, i.e., we are simultaneously doing CM and ATS. **e** Linear actuator to

mimic ATS for conducting controlled experiments. The gap between the battery and heat sink can be tuned as the actuator contracts or elongates and, thus, the thermal contact changes. **f** Representative battery temperature evolution for XFC experiments using 5-Ah C‖LCO pouch cells with different thermal protocols indicated by the line colors. The line styles denote four phases of the protocol, starting with XFC at 6 C (16.98 mA/cm²). **g** Effective $h$ between the battery and coolant for various cases obtained by matching the thermal model with the experimental temperature data from (**f**) (Supplementary Fig. 5). $h$ with the switch OFF and ON was comparable to $h$ for insulation and cooling protocols, respectively. Note that $h_{OFF}$ for CM alone is much higher than that for switch + CM resulting in a significantly lower $T_B$ rise (Fig. 1f).

can lower the overall performance and lifetime[25–27] (Supplementary Fig. 2). This effect becomes noticeable with increasing ambient temperature ($T_a$), as the side reaction rates increase rapidly with increasing temperature. Further, adding extra metal foils in the cell is incompatible with the existing battery manufacturing process[25], which has been perfected by the industry over several decades. The intrusive nature of the embedded heater may also raise safety concerns. Thus, neither of these heating strategies can enable XFC operation for variable $T_a$ and manufacturing and safety concerns.

In this work, we propose a thermally modulated charging protocol (TMCP) that combines the desired traits of both the cell- and system-level strategies via active thermal switching (ATS), i.e., low $h$ ($h_{OFF}$) at the cell level during XFC and high $h$ ($h_{ON}$) during discharge/rest ("Switch + CM" in Fig. 1a). Benefiting from the efficient use of the battery's self-generated heat, our approach does not rely on extra heat sources such as the embedded heaters proposed by Yang et al.[20,21,23], and thus is completely nonintrusive. A proof-of-concept study of commercial high-energy-density C‖LCO LIBs under various thermal protocols demonstrates that our TMCP consistently outperforms existing thermal protocols. By implementing the TMCP, the XFC performance of commercial LIBs exceeds the key US DOE targets. Electrochemical analysis and postmortem characterization using optical microscopy, scanning electron microscopy (SEM), and X-ray tomography suggests that the improved XFC performance can be attributed to the mitigated lithium plating during XFC and reduced side reactions during discharging by ATS. For practical implementation of our approach with existing BTMSs, we developed an ATS device with small mass and volume (1.4% and 3.0% compared with that of a battery, respectively) using a cost-effective shape memory alloy, which has the potential to enable XFC in commercial battery packs.

## Results and discussion
### TMCP design and validation

To design the TMCP, we evaluated the effect of $h$ on the XFC of representative C‖LCO cells using our ECT model (Supplementary Fig. 1 and Supplementary Note 1). The simulation indicated the range of $h_{OFF}$ needed to reach optimal $T_B$ (~45 °C) and avoid negative electrode potentials below 0 V vs. Li/Li$^+$ for mitigation of lithium plating during XFC (Supplementary Fig. 3). Since the timescale for XFC is 10–15 min, a high initial temperature (e.g., 50 °C) is not sufficient for maintaining the high battery temperature (>45 °C) during the whole charging process if the battery is not appropriately thermally insulated (Fig. 1b, c). Figure 1d obtained from our ECT model reveals that a switching ratio ($h_{ON}/h_{OFF}$) of ~10 is needed as high $h_{ON}$ is desired for the other states (rest and discharging). As a proof of concept, we performed 6C1C cycling tests (6C charge and 1C discharge; see "Methods" for details) of commercial high-energy-density LIBs with our TMCP. Representative 5-Ah C‖LCO LIBs with a specific energy of 240.8 Wh/kg at C/3, the maximum recommended charge rate of 1C, and the maximum discharge rate of 2C were used for this XFC study (see details in "Methods"). A linear actuator was used to simulate ATS experimentally to validate the efficacy of TMCP (Fig. 1e and Supplementary Fig. 4). The springs hold the battery away from the heat sink with an air gap during XFC (switch OFF and coolant flow off), while the actuator elongates after XFC and pushes the battery in contact with the heat sink (switch ON and coolant flow on). Apart from ATS, controlled experiments were conducted using other thermal protocols for XFC (Fig. 1a): (1) cooling: the coolant flow was always on (i.e., the conventional thermal scheme[28]); (2) coolant modulation (CM OFF/ON): the coolant flow was off during XFC and on during resting and discharging (i.e., the system-level strategy[24]); and (3) insulation: the cell was always thermally insulated, as proposed by Yang et al.[21].

Figure 1f displays the representative battery-temperature evolution in an XFC cycle under different thermal protocols. With the OFF state for switch and CM, the rise of $T_B$ during XFC was comparable to that in the insulation case. Note that turning off the coolant flow only (CM OFF) cannot raise $T_B$ significantly due to the heat leakage from the battery to the heat sink. After XFC, turning on the switch allows for efficient cooling and optimal control of $T_B$. In contrast, the insulation case led to high temperature during rest and discharge, which is highly undesirable[27] (Supplementary Fig. 2). By fitting the temperature profile with thermal models (Supplementary Fig. 5 and Supplementary Note 2), the effective $h$ was extracted for various cases (Fig. 1g). The results indicate that the switching ratio of 10.4 is >10, as required (Fig. 1d), and that the effective $h$ with the switch OFF and ON was comparable to $h$ for insulation and cooling, respectively. For comparison, the effective $h$ with coolant OFF state in CM is ~3 times that with the OFF state of the switch and CM (proposed strategy).

For this type of cell (5-Ah C‖LCO LIBs), the charge time ($t_c$) to 80% state of charge (SOC) depends highly on the thermal protocol and temperature rise (Fig. 2a and Supplementary Fig. 6). The charge time with the coolant flow on (cooling) and off (representing OFF state in CM) was ~25 and ~18 min, respectively. It decreased to <15 min for insulation and switch + CM protocols due to the boost of the battery kinetics by high temperature. As discussed earlier, the high temperature is beneficial to avoid negative electrode potentials below 0 V vs. Li/Li$^+$ and mitigate Li plating during XFC, which can be verified by comparing the coulombic efficiency (CE) of these cases (Fig. 2b). The low or high CE indicates the presence or mitigation of lithium plating[9]. The high CE in the insulation and switch case further led to the extended 6C1C cycle life associated with 20% capacity fade (Fig. 2c). The lower cycle life in the insulation case (665 cycles), compared with the switch case (975 cycles), is attributed to the increased side reaction rates at higher discharge temperature (35–40 °C). This effect is more pronounced at higher ambient temperatures (~40 °C), resulting in a cycle life of 334 cycles for the insulation compared with that of 560 cycles for the switch+CM, which still exceeds the US DOE target (Fig. 2d, e). Figure 2f shows the XFC cycle life, which is defined as the number of cycles that batteries can be charged to 80% SOC in 15 min. Because of the limit of $t_c$ (Supplementary Fig. 6), the XFC cycle life can be lower than the overall cycle life at 20% capacity loss (Fig. 2b); e.g., the XFC cycle life for the case of cooling and CM is 0. Our approach enables >500 XFC cycles at ambient temperatures near or above typical room temperature (≥25 °C), and the advantage of "switch" over "insulation" increases with ambient temperature. In the insulation case, the cell had to rest for an extra 15 min before starting the next charge cycle as compared to the switch, as it takes longer for the thermally insulated cell to cool down to the ambient temperature (Figs. 1f and 2d). If the extra 15 min of rest is not provided for the thermally insulated cell, the expected XFC cycle life will be significantly lower due to an even higher average temperature.

The capacity loss behavior shown in Fig. 2c arises from a complex combination of Li plating and side reactions such as solid electrolyte interphase (SEI) growth[29–31]. To better understand the effects of switch and cooling on the capacity fade (i.e., our TMCP and the conventional thermal scheme for XFC), we investigated the degradation mechanism with electrochemical analysis (Supplementary Fig. 7) and postmortem characterization via optical microscopy, SEM, and X-ray tomography (Fig. 3; see Fig. 3a, d for the images of the uncycled negative electrode). For the cooling protocol, a large portion of the aged negative electrode is covered by plated Li, and the individual particle features become hardly visible due to the coverage (Fig. 3b, e). By comparison, the particle features remain visible in most parts of the aged negative electrode for the switch, while some particles are covered by a layer of reaction products (Fig. 3c, f). These observations confirm the presence (cooling) or mitigation (switch) of lithium plating as revealed by the CE analysis (Fig. 2b). This explains the different rates of capacity fade in the initial linear aging regime, which speeds (cooling) or delays (switch) the transition to the rapid nonlinear capacity fade stage and hence largely determines the cycle life. In contrast, at the end of life,

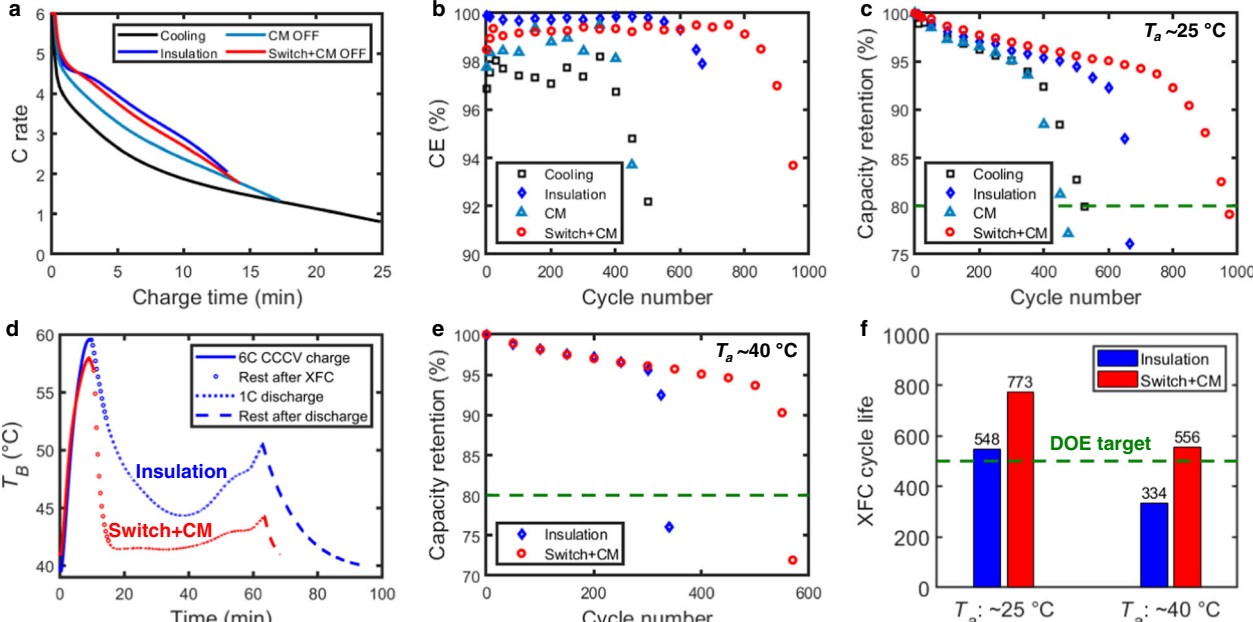

**Fig. 2 | XFC cycling results. a** Evolution of C rate during 6C constant-current constant-voltage charging to 80% SOC. **b** Evolution of coulombic efficiency (CE) with cycle number. The low CE in the case of cooling and CM is attributed to the severe lithium plating related to XFC. The increased CE in the insulation and switch case is due to the mitigation of lithium plating at high temperatures during XFC. **c** Capacity retention of the cells cycled at $T_a$ of -25 °C with different thermal protocols. **d** Representative temperature evolution for the cells tested at -40 °C

ambient temperature for insulation and switch. Note in the case of thermal insulation, the cell rested for an extra 15 min after discharge before starting the next charge cycle as compared to the switch, as it takes longer for the thermally insulated cell to reach the ambient temperature (see text for details). This was also the case for the 25 °C ambient temperature (Fig. 1f). **e** Capacity retention of the cells cycled at $T_a$ of -40 °C. $T_B > 45$ °C for a longer duration led to cycle life <500 for the insulation case. **f** XFC cycle life of the cells tested at $T_a$ of -25 and -40 °C.

the switch case experiences a greater loss of negative electrode porosity than the cooling case, as quantified with tomography (Fig. 3g–i). The higher porosity loss in the aged negative electrode from the switch case is, in part, a result of more SEI growth related to the longer cycle life and higher operation temperature. This agrees with the larger SEI resistance observed in the aged cell from the switch (Supplementary Fig. 7 and Supplementary Table 1). Finally, we confirmed that the loss of active materials or change of the positive electrode has a limited impact on the observed capacity degradation (Supplementary Table 2 and Supplementary Figs. 8 and 9).

**Prototype device development and demonstration**

This proof-of-concept study clearly demonstrates the benefits of TMCP for XFC. To integrate our approach into existing BTMSs, we recommend and abided by the following design rules for ATS devices: (1) switching ratio ≥10; (2) minimal impact on the system-level specific energy and energy density; (3) small power consumption for switching; (4) zero power consumption in the ON/OFF state to maximize the energy efficiency; (5) compatibility with existing BTMSs. For the switching ratio ≥10, we selected a solid/solid mechanical switch based on contact and separation as the most promising option based on the thorough review of thermal switches by Wehmeyer et al.[32]. Thus, we built our prototype device by integrating a mechanical thermal switch based on a shape memory alloy (SMA) with a heat-sink plate (Fig. 4a).

SMA wires were selected for the active actuation using temperature-responsive phase and volume change, which satisfies the traits mentioned above. We selected Nitinol SMA wires with a phase-transition temperature >60 °C to avoid any passive response to the environmental temperature. A bistable structure consisting of one spring steel strip and two pivot blocks was used for energy savings, i.e., energy is only consumed for the change of state (Fig. 4a). The strip becomes concave or convex depending on the contraction of the two SMA wires, which trigger the rotation of the pivot blocks. To switch

from OFF to ON, a current pulse (1.5 A and 5 s) is applied to the right SMA wire to trigger the wire contraction and rotate the corresponding pivot block anticlockwise, and the gap between the battery and heat-sink plate is closed as the spring strip becomes concave (Supplementary Fig. 10a). In the ON state, the pressure between the battery and heat sink comes from the elastic bands, which are used to mimic the pressure in practical battery pack assemblies. Similarly, heating the left SMA wire rotates the left pivot block anticlockwise, converting the spring strip to convex, thereby overcoming the elastic band force to lift the battery pack and open a thermally insulating air gap (-0.5 mm), thereby bringing the state back to OFF (Supplementary Fig. 10b, c). The electrical energy consumed for one such change of thermal state is -0.01 Wh per cycle, which is negligible (-0.05%) compared with the charge/discharge energy of the cell (-18.5 Wh).

Using this device, we performed the XFC cycling test of 5-Ah C||LCO cells with the proposed TMCP. Infrared thermal images of the cell display the different battery surface temperatures during XFC (42.2 °C) and discharging (30.4 °C) (Fig. 4a). ATS by the SMA device and the linear actuator demonstrated comparable thermal switching capability and resulted in similar evolution of temperature (Fig. 4b, c rate (Fig. 4c) in an XFC cycle. Regardless of the ATS method, our TMCP led to the XFC performance exceeding the targets set by DOE in terms of charge time (Supplementary Fig. 11), capacity retention (Fig. 4d), and discharge specific energy after XFC (Fig. 4e). The XFC cycle life (773 cycles) exceeded the DOE target (500 cycles) by 54.6%, and the C/3 discharge specific energy after XFC was higher than the DOE target over the entire XFC cycle life. Further, we compared our approach with the method proposed by Yang et al. using a heater-embedded battery structure[21]. Both our and their approach beat the DOE target for cycle life (Fig. 4d); however, our commercial high-energy-density cell also achieves the DOE specific energy target as compared to Yang et al. (Fig. 4e). Considering the complex battery-to-battery differences (i.e., our commercial C||LCO cells and their home-built C||Lithium nickel manganese cobalt oxides cells), it is unreasonable to directly compare

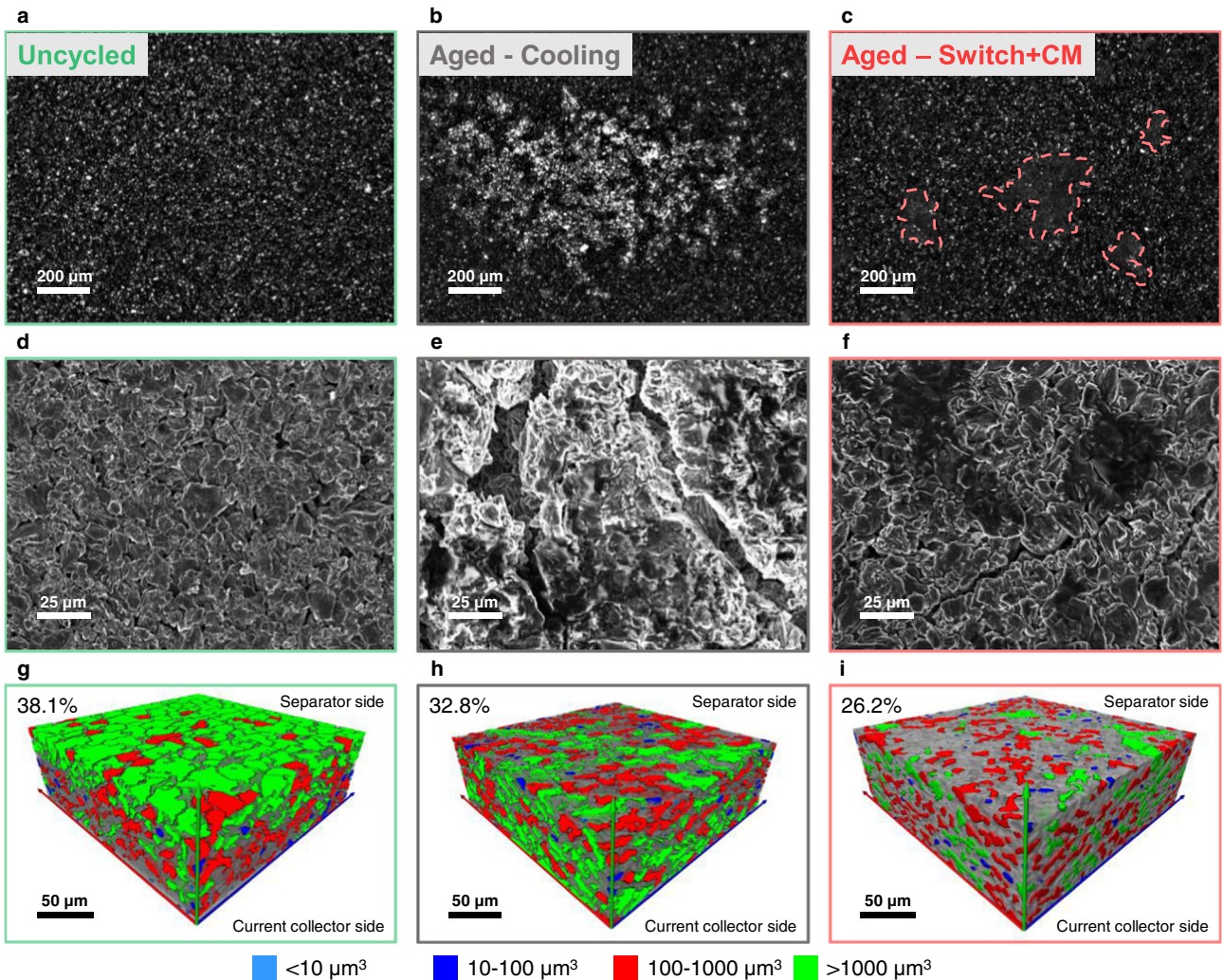

**Fig. 3 | Degradation mechanisms.** Representative **a**–**c** optical images, **d**–**f** SEM images, and **g**–**i** X-ray tomography images of uncycled and aged negative electrodes for cooling and switch, respectively (see text for discussion). From the tomography images, the reduction of pore size on the negative electrode surface (e.g., fewer large pores with volume >1000 μm³) indicates pore clogging.

the total cycle life (N) and the total operation time (t) at 20% capacity loss. Instead, the benefit of these two methods for XFC was evaluated using a ratio normalized by N or t from low-rate cycling tests where side reactions dominate the capacity fade (e.g., 1C charge and 1C discharge; Supplementary Fig. 12), i.e., $N_{6C1C}/N_{1C1C}$ or $t_{6C1C}/t_{1C1C}$ (Fig. 4f). With our TMCP, the $t_{6C1C}/t_{1C1C}$ of 96.2% demonstrated that the XFC induced degradation due to plating is largely reduced without apparently accelerating side reaction rates. In contrast, this ratio is 44.3% in Yang et al.'s work due to the increased side reactions related to the thermal insulation and the high operation temperature (~60 °C). Thus, our strategy fully exploits the potential of batteries for XFC with comparable operation time of 6C1C and 1C1C cycling.

In addition to the XFC performance, the relative mass, volume, and material cost of the SMA-based thermal switch, compared with that of a battery, are estimated to be 1.4%, 3.0%, and 0.7%–1.9%, respectively (Supplementary Note 3, Supplementary Tables 3 and 4). The mass, volume, and cost of this BTMS-integrated switch will be even lower for higher-capacity cells and can be further reduced by optimizing the design in future work. Considering the long R&D and commercialization cycles (~15–20 years) for new electrolytes and electrode materials, our approach using existing cost-effective materials (i.e., SMA wires) could provide a short- to medium-term solution for enabling XFC.

## Application of the XFC method in various Li-ion cell chemistries and operating conditions

So far, the efficacy of our approach is evaluated based on the XFC cycling tests of commercial high-energy-density C‖LCO cells from SOC = 0 to 80% at $T_a$ of 25 and 40 °C. In comparison, most XFC studies in the literature were performed with lower energy-density LIBs, from SOC = 0 and at $T_a$ = 25–30 °C[13–18,21,24]. Since the boost of battery kinetics at elevated temperatures is universal, the benefit is not limited by the type of positive electrode, the initial SOC, and the initial temperature. For XFC of graphite negative electrode-based LIBs, Li plating on graphite negative electrodes is known as the dominant degradation mechanism[2–4]. The mitigation of Li plating with our approach is universally applicable to mainstream EV LIBs using lithium nickel manganese cobalt oxides (NMC) or lithium iron phosphate positive electrode active materials. The high temperature rise during XFC may accelerate positive electrode aging, especially in C‖NMC cells. From our XFC study of commercial C‖NMC cells, this effect is minor compared to the benefit of Li plating mitigation (Supplementary Fig. 13), which originates from the minimized exposure time to high temperature by thermal switching after XFC, i.e., during rest and discharge.

As for the effect of $T_a$ and initial SOC, the likelihood of Li plating increases at low $T_a$ and high SOCs, which highlights the

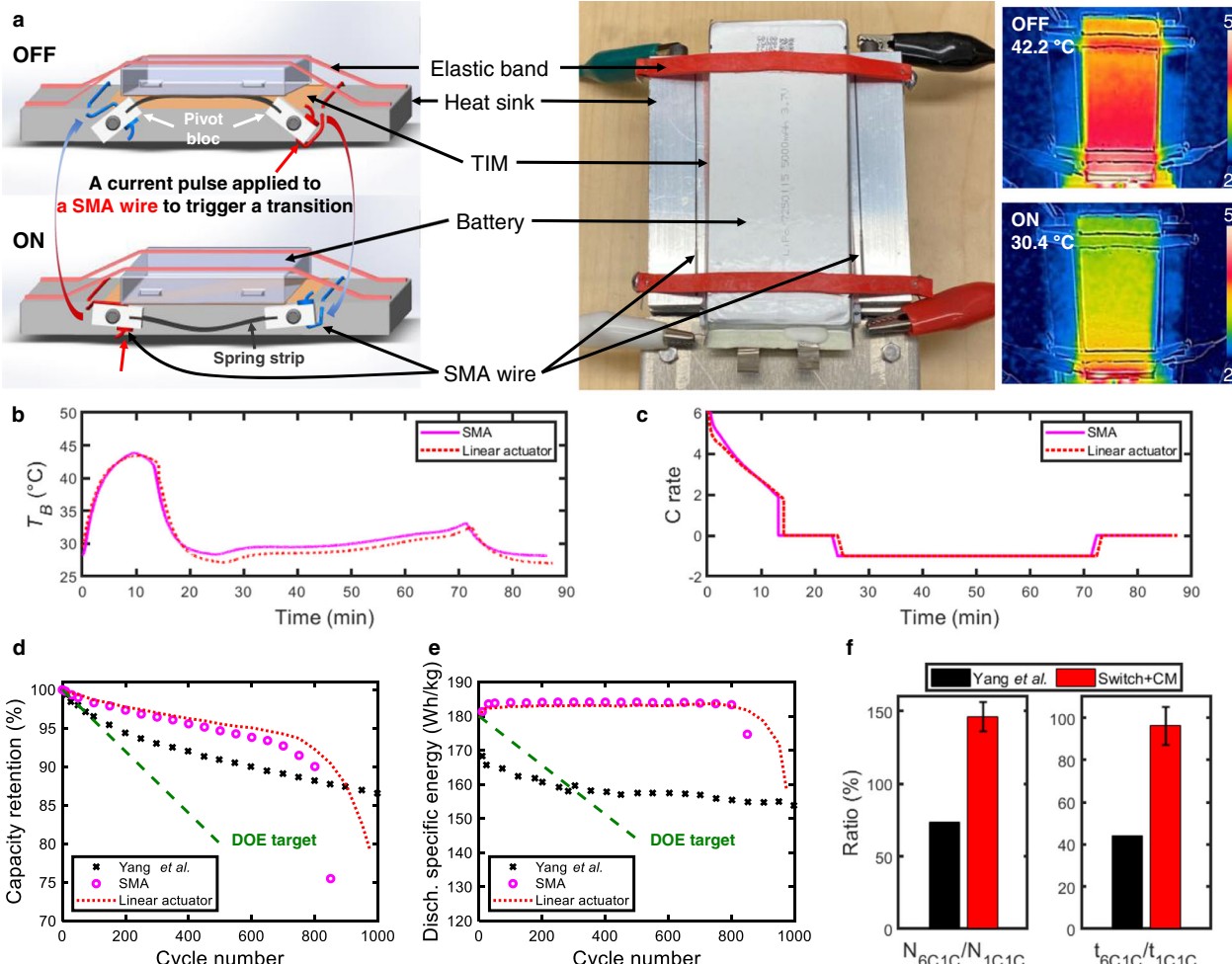

**Fig. 4 | Performance of SMA-based thermal switch integrated with a BTMS.**
**a** Schematic, photograph, and infrared thermal image of the SMA thermal switch. The state of thermal contact is toggled by briefly heating one of the two SMA wires with a current pulse, i.e., the heated SMA wire contracts and triggers the switch to either open (left wire) or close (right wire). Representative evolution of **b** temperature and **c** C rate in an XFC cycle for 5-Ah C‖LCO cells by thermal switching. Comparison of **d** capacity retention and **e** discharge specific energy with the XFC target set by US DOE (the green dashed line in (**d**, **e**), i.e., <20% capacity loss in 500 XFC cycles and >180 Wh/kg discharge specific energy after XFC, respectively. **f** A comparison of normalized performance ($N_{6C1C}/N_{1C1C}$ and $t_{6C1C}/t_{1C1C}$) between our approach and the method by Yang et al.[21].

need for boosted battery kinetics using battery heat. To validate this hypothesis, we performed XFC tests from a representative nonzero initial SOC (5–15%)[33] (Supplementary Fig. 14) and/or at low $T_a$ (Supplementary Fig. 15). The benefit of our method is significant in both scenarios. At low $T_a$ (e.g., −20 °C), fast charging is particularly difficult given the challenge of XFC, even at 25 °C. Our thermal switching strategy alone cannot achieve XFC at such low $T_a$. We anticipate that realizing such a goal requires extensive multilevel research and development, including thermal management. Note that most previous XFC studies were performed at $T_a = 25$–30 °C and from SOC = 0[13–18,21,24]. In fact, the primary metric for low-temperature operation is that the discharge energy at −20 °C should be >70% of that at 30 °C, with C/3 charging at 30 °C[7]. Active thermal switching can improve discharge performance at low temperatures by retaining battery heat as needed[34]. With our approach, the relative discharge energy at C/3 and 1C discharge rate compared to that at 30 °C is 78.8% and 85.0% (Supplementary Fig. 16), respectively. At both discharge rates, the relative discharge energy is higher than the 70% target set by the United States Advanced Battery Consortium. Further, retaining the heat during discharging is beneficial to recharging at low temperatures (Supplementary Figs. 15 and 17).

## System-level consideration

Our approach does not rely on the modification of single cells, and the nonintrusive nature is advantageous for the application at different scales. For high-capacity cells, the size of our device can increase accordingly as the cell size increases (see the test of 60-Ah cells in Supplementary Fig. 18), and the benefit of "switch" over "insulation" further increases due to the increased cooling demand for the thick cells during discharging. At the pack level, the contact and separation between the cells and the cold plate can be controlled in a similar manner, e.g., moving the cold plate via the contraction of SMA wires (see Fig. 5 for a potential pack-level design). Like the device-level switch, this proposed pack-level thermal switch only operates between the battery pack and the cold plate and thus does not affect the heat transfer between batteries. The relative volume, mass, and material cost of the pack-level switch are estimated to be comparable to that of the device-level thermal switch, which shows the promise of practical applications. Further, we speculate that our switch can be adapted to batteries and packs of different sizes due to the nonintrusive nature. Depending on the battery geometry, the front or side surface is selected as the contact surface for cooling/heating in commercial battery packs. Our thermal switch can achieve contact or separation regardless of the surface used for heat transfer. The effectiveness of

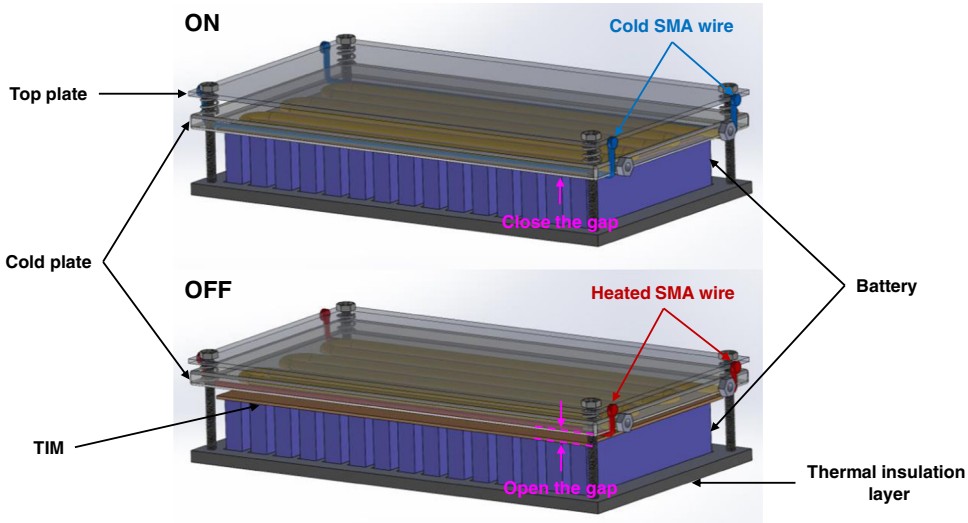

**Fig. 5 | A potential pack-level design for active thermal switching.** At the pack or module level, the gap between the battery and the cold plate can be controlled by moving the cold plate by heating the SMA wires. To switch from ON to OFF, the self-heated SMA wires contract and open the gap between the cold plate and the battery. The application of our approach to the pack level requires minor modifications, which benefits from the nonintrusive nature. Parts in the schematic are not to scale and can be adapted to batteries of different sizes.

our strategy for thermal management using the side surface of batteries is verified by ECT simulations, as shown in Supplementary Fig. 19.

In addition, the thermal switching strategy is universal to various BTMSs. Here, our SMA-based thermal switch is developed for decoupling the thermal mass of the battery and the cold plate (or heat sink) in the mainstream liquid cooling system. Note that liquid cooling is used in most EVs for many reasons, such as pack-level temperature uniformity. For other BTMSs, e.g., air cooling systems, thermal switching can be potentially realized by controlling the airflow velocity.

In summary, we have developed a thermal solution for enabling XFC in commercial high-energy-density LIBs. Unlike previous innovations in battery materials, our approach leverages battery intrinsic heat to improve the XFC performance using a BTMS-integrated thermal switch based on existing cost-effective materials. Considering the dependence of the optimal system temperature on the operating condition, the optimal temperature can be continuously adjusted depending on the condition of the cell via the ATS in a smart BTMS.

## Methods
### ECT simulation and verification
We coupled the Lithium-Ion Battery Module and the Heat Transfer Module in COMSOL Multiphysics 5.6 for the simulation of battery operation in different thermal conditions based on a Newman pseudo 2D electrochemical model and a 3D transient heat transfer model (Supplementary Note 1). As a verification of the ECT model in predicting the negative electrode potential, we assembled three-electrode cells (i.e., 32 mAh C||LCO single-layer pouch cells using lithium foil as the reference electrode) in an argon-filled glovebox (Nexus II, VAC; $O_2$ < 0.5 p.p.m. and $H_2O$ < 0.5 p.p.m) for measuring the negative electrode potential (Supplementary Fig. 1). The lithium metal reference electrode (MSE Supplies, USA; Width × Length × Thickness: 0.1 cm × 0.3 cm × 0.2 mm; ≥99.9% purity) was placed between the double separator layers. We used the negative and positive electrodes from Argonne's Cell Analysis, Modeling and Prototyping (CAMP) Facility. The material specifications associated with the three-electrode cells can be found in Supplementary Table 5.

### XFC cycling experiments
Commercial high-energy-density 5-Ah C||LCO LIBs (model number: PL-7250115-2C) were used in this work. 1C is the current needed to fully charge or discharge the nominal capacity (5 Ah) in 1 h. The nominal specific energy and calibrated C/3 specific energy are 205.5 and 240.8 Wh/kg, respectively. From the manufacturer[35], the recommended maximum charge rate is 1C for such high-energy-density pouch cells. While the rate capacity is better in low-energy-density cells, these cells are not selected as their discharge energy density is much lower than the DOE target. The charge cutoff voltage is 4.2 V, and the discharge cutoff voltage is 3.0 V.

In the XFC cycling tests, batteries are charged at 6C to 80% SOC using a standard constant-current constant-voltage (CCCV) charging protocol with a cutoff voltage of 4.2 V. After a 10-min rest, batteries are discharged at 1C to the cutoff voltage specified by the manufacturer. The rest time after discharge is 30 min for the case of insulation and 15 min for other cases, which ensures thermal equilibrium at the end of each XFC cycle, e.g., <1 °C temperature difference compared with the ambient temperature. After each 50 XFC cycles, the capacity is calibrated by charging and discharging at C/3, and then, the discharge energy density after XFC is quantified by 6C CCCV charging to 80% SOC and discharging at C/3. For each condition, the test is repeated at least twice, and the relative deviation of cycle life is ~5% (4.2% for "cooling"; 4.8% for "CM"; 5.2% for "Insulation"; 5.4% for "CM + Switch").

All the cells were tested using an 8-channel Arbin Laboratory Battery Testing System (LBT21084). Tests above 0 °C were performed in a TestEquity thermoelectric temperature chamber (TEC1), and the test at −20 °C was done in a TestEquity temperature chamber (model 107), respectively. The cycler was calibrated by the manufacturer before use. The TEC1 chamber was off for the XFC cycling tests at -25 °C to mimic real-world scenarios and was set to 40 °C to simulate higher ambient temperature.

### Thermal switch experiments
For the conceptual thermal switch, a linear actuator (L12-R micro linear servos for RC & Arduino, Actuonix) was used and controlled using a microcontroller board (Arduino UNO). The board acquires the battery operation status (i.e., charge or discharge) from the derivative of the cell voltage with respect to time and changes the status of the actuator (i.e., the original or elongated state) accordingly. The same logic was

applied to the thermal switch device based on SMA. At the start or end of XFC, the status of the switch is tuned by heating a specific SMA wire (0.254 mm Nitinol wire, Kellogg) using a current pulse. The thermal interface material (TIM) layer (Laird TPLI210) was selected to ensure good thermal contact and durability. Infrared images were taken using a thermal imaging camera (FLIR E4).

## Postmortem characterization

The uncycled and aged 5-Ah C||LCO cells were fully discharged at C/3 to 2.75 V and then disassembled in an Ar-filled glovebox ($O_2 < 0.1$ p.p.m. and $H_2O < 0.1$ p.p.m.). The graphite negative electrodes were sealed in a chamber to ensure limited exposure to oxygen or moisture during optical characterization using a confocal microscope (Lasertec L7 Hybrid). For SEM imaging and tomography, the electrode samples were washed multiple times with ethylmethyl carbonated solvent and dried in a vacuum chamber before transfer to the instrument. SEM images were taken using an FEI Quanta 3D FEG FIB/SEM. X-ray tomography (24 keV X-ray; 180° rotation with a 0.072° step) was conducted at Beamline 8.3.2 at the Advanced Light Source (ALS) at Lawrence Berkeley National Laboratory. With the X-ray shutter OFF, we collected the dark field images for the deduction of detector dark counts. With the X-ray shutter ON, the bright field images taken before and after the sample scan were used to normalize the variations of the incident illumination. We performed the 3D reconstructions with TomoPy and visualized the reconstructed slices with Avizo software[36].

## Electrochemical impedance spectroscopy

The electrochemical impedance spectroscopy (EIS) measurements were performed using a VMP3 multichannel potentiostat. The cells were discharged at 1C to 3.0 V and held at this open-circuit voltage (~3.42 V) for ~2 h before the EIS tests. A signal with 50 mA amplitude was applied to the cell with a frequency ranging from 0.05 to 10 kHz and 15 data points per decade of frequency. An equivalent circuit model[37] was used to fit the EIS results (Supplementary Note 4 and Supplementary Fig. 7). The parameters determined from the EIS analysis were summarized in Supplementary Table 1.

## Reporting summary

Further information on research design is available in the Nature Portfolio Reporting Summary linked to this article.

# Data availability

The data supporting the findings of this study are available from the corresponding author upon reasonable request.

# Code availability

The code used in this study is available from the corresponding author upon reasonable request.

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

## Acknowledgements

The authors acknowledge the support of Energy Efficiency and Renewable Energy, Vehicle Technologies Program, of the US Department of Energy under contract no. DEAC0205CH11231. This work used Beamline 8.3.2 at ALS, a DOE Office of Science User Facility under contract no. DEAC0205CH11231.

## Author contributions

Y.Z. and R.S.P. conceived the idea. Y.Z. and B.Z. designed and conducted the electrochemical simulations and cycling experiments. Y.Z., B.Z., Y.F., F.S., Q.Z., and D.C. conducted the postmortem characterizations. Y.Z., R.M., and R.S.P. designed the bistable thermal-switch device. S.K., S.D.L., M.C.T., V.B., and C.D. discussed the results. Y.Z., B.Z., and R.S.P. contributed to the writing of the manuscript.

## Competing interests

The authors declare no competing interests.
