## [Peer Review File · Nature Communications]

REVIEWER COMMENTS

Reviewer #1 (Remarks to the Author):

The concept and entire work presented in this manuscript is premised on the equation stated on Line 44, which is fundamentally wrong. This equation appears to violate the principle of energy conservation and does not include heat capacitive effect. Even with zero heat transfer coefficient or surface area, the battery temperature would not go to infinity as suggested by this equation. Battery thermal management is a transient heat transfer problem, not at steady-state as the equation implies. I would think that transient effects are dominating in the application examined in this paper; if the authors disagree, they should present a detailed analysis of why transient effects can be neglected in the equation.

In transient heat transfer, thermal boundary conditions are not that important, as the heat loss equals to multiplication of heat transfer coefficient, surface area, temperature difference, and the time. If the time is short as in this application, the total heat loss is small regardless of heat transfer coefficient (i.e. fast transient heat transfer).

If preheating time is so short, say less than 1 minute, why does the battery need to be thermally insulated (Line 62). I cannot understand this from the principles of heat transfer. I suggest that the authors bring in additional heat transfer experts into the team to fully analyze thermal energy balance and clear fundamental mistakes before this work can be published.

Reviewer #2 (Remarks to the Author):

The active thermal switching structure can retain the heat during XFC with the switch OFF to boost the kinetics while dissipating the heat after XFC with the switch ON to reduce side reactions. There are still some issues that need to be addressed in the application of battery thermal management.

1. How is the active switch increases the battery's temperature during charging? In my opinion, the active switch is used to control the heat dissipation to adjust the battery temperature passively.
2. If the battery is in a low temperature for a long time, can the active switch increase the battery's temperature?
3. From a practical standpoint, can this active switch structure used in a battery pack? The extra gap volume affects the heat transfer between batteries and it is not permitted in a battery pack.
4. Fig.S19 shows the pack-level design for active thermal switching. However, the heat dissipation surface of the pack-level is the side-surface of the batteries, while that of single battery is the front-surface of the battery. Does the active thermal switch still have good thermal control performance when the side-surface of the battery is in contact with the cold plate?

We thank the Reviewers for their time and appreciate their valuable suggestions to improve the manuscript. Here we provide a detailed point-by-point response to the Reviewers' comments, and we have edited the manuscript and Supplementary Information accordingly.

Color codes used in this response letter:

Black Italic: original review comments;

Blue: our responses;

Red: revisions made in the manuscript.

Reviewer #1 (Remarks to the Author):

The concept and entire work presented in this manuscript is premised on the equation stated on Line 44, which is fundamentally wrong. This equation appears to violate the principle of energy conservation and does not include heat capacitive effect. Even with zero heat transfer coefficient or surface area, the battery temperature would not go to infinity as suggested by this equation. Battery thermal management is a transient heat transfer problem, not at steady-state as the equation implies. I would think that transient effects are dominating in the application examined in this paper; if the authors disagree, they should present a detailed analysis of why transient effects can be neglected in the equation.

Response:

We would like to clarify that our work is based on transient heat transfer analysis as detailed in Supplementary Note 1. The transient temperature profile during XFC, as shown in Supplementary Figure 1 d and e, is calculated using our transient model. To make this clear, the governing equations of transient heat conduction used in our model are added in the revised Supplementary Note 1.

We speculate that the confusion could come from the approximate format of the expression on Line 44, *i.e.*, $T_B \approx Q / hA + T_C$. **This expression was used only to explain that T_B has a positive correlation with Q , $1/hA$, and T_C . We never used this equation to model/design the battery+ thermal switch system.** The corresponding transient heat conduction equation for a lumped system (*i.e.*, ignoring the heat diffusion due to finite thermal conductivity and size of the battery, which is a very good assumption for low values of heat transfer coefficient as discussed below and in the revised paper) is $Q(t) = mC_p \frac{\partial T_B}{\partial t} + hA(T_B - T_C)$, where m and C_p are the mass and heat capacity of battery, respectively. It can be rewritten as $T_B(t) = (Q(t) - mC_p \frac{\partial T_B}{\partial t}) / hA + T_C$. Note that $Q - mC_p \frac{\partial T_B}{\partial t} = hA(T_B - T_C)$ can only be positive for raising the battery temperature above T_C (*i.e.*, $T_B - T_C > 0$). Since both Q and $Q - mC_p \frac{\partial T_B}{\partial t}$ are positive, the battery temperature T_B increases with Q/hA and T_C . In the revised manuscript, we make this clear and refer to Supplementary Note 1 for details on the transient electrochemical-thermal model.

Further, we performed 3D transient heat transfer simulations including heat diffusion due to finite thermal conductivity and size of the battery for all the cases and updated the simulation results accordingly in the revised version. The results also verified the accuracy of the previously used lumped thermal model in the range of low heat transfer coefficient needed for XFC. This is anticipated as the lumped-capacitance model is a common approximation in transient heat

conduction analysis and is effective when the heat conduction within an object is much faster than the heat convection on the boundary (e.g., our case with low heat transfer coefficient).

Revision:

In the revised manuscript, we have removed the original $T_B \approx Q/hA + T_c$ equation and replaced it with the transient equation. We clarify the positive correlation of T_B with $Q/(hA)$ and T_c , and refer to Supplementary Note 1 for details on the 3D transient electrochemical-thermal model (Page 2 and 3):

“Treating the battery as a lumped thermal system (see Supplementary Note 1 for details on the validity of lumped model), the transient battery temperature can be written as $T_B(t) = (Q - mC_p \frac{\partial T_B}{\partial t})/hA + T_c$, where Q , A , T_B , m , C_p are the transient heat generation, surface area, temperature, mass, heat capacity of the battery, respectively. T_c is the coolant temperature and h denotes the tunable thermal conductance per unit area between the battery and coolant, as illustrated in Fig. 1a. Thus, elevating the battery temperature relies on the increased Q and T_c and/or the reduced hA (see Supplementary Note 1 for the 3D transient electrochemical-thermal model).”

Full 3D transient heat transfer simulations including heat diffusion are performed as described in the revised Supplementary Note 1 and the simulation results in Supplementary Figure 1 and 3 are updated accordingly. Note that the desired low h obtained from both the lumped-capacitance model and 3D simulations are same because lumped capacitance model is a good assumption for low Biot numbers given by hd/k (Reference: Incropera, Frank P.; DeWitt, David P.; Bergman, Theodore L.; Lavine, Adrienne S. (2007). Fundamentals of Heat and Mass Transfer (6th ed.). John Wiley & Sons. pp. 260–261. ISBN 978-0-471-45728-2. OCLC 288958608), where d is the thickness of the battery and k is the thermal conductivity. As shown in the representative 3D temperature distribution, the difference between the battery temperature and average temperature is within ~ 0.5 °C for $h = 10$ W/m²K, and thus is negligible compared to the absolute temperature rise as obtained from the lumped model. Therefore, the desired h range shown in the manuscript does not change.

Supplementary Note 1: Electrochemical-thermal (ECT) simulation

Electrochemical-thermal (ECT) simulations were performed in COMSOL Multiphysics 5.6. We coupled the Lithium-Ion Battery Module and the Heat Transfer Module for the simulation of battery operation in different thermal conditions, based on a Newman pseudo 2D electrochemical model and a 3D transient heat transfer model.

The following conservation equations are solved to describe the 1D electrochemical processes through the porous electrode along with the Li diffusion inside the active material particles.

Charge conservation in solid particles:

$$\nabla \cdot (\sigma_s^{eff} \nabla \phi_s) - j = 0. \quad (S1)$$

Charge conservation in electrolyte:

$$\nabla \cdot (\kappa^{eff} \nabla \phi_e + \kappa_D^{eff} \nabla \ln c_e) + j = 0, \quad (S2)$$

in which the effective diffusional ionic conductivity:

$$\kappa_D^{eff} = \frac{2RT\kappa^{eff}}{F} (t^+ - 1) \left(1 + \frac{d \ln f_{\pm}}{d \ln c_e}\right). \quad (S3)$$

Material conservation in electrolyte:

$$\varepsilon \frac{\partial c_e}{\partial t} = \nabla \cdot (D_e^{eff} \nabla c_e) + \frac{1-t^+}{F} j. \quad (S4)$$

Material conservation in solid particles is governed by the Fick's law in sphere:

$$\frac{\partial c_s}{\partial t} = \frac{1}{r^2} \frac{\partial}{\partial r} (D_s r^2 \frac{\partial c_s}{\partial r}), \quad (S5)$$

with boundary condition on particle surface:

$$-D_{s,i} \frac{\partial c_{s,i}}{\partial r} |_{r=R_i} = \frac{i}{F}. \quad (S6)$$

Butler-Volmer equation for charge transfer kinetics relating the reaction current (i) with the surface overpotential:

$$i = i_0 [\exp(\frac{\alpha_a F}{RT} \eta) - \exp(-\frac{\alpha_c F}{RT} \eta)], \quad (S7)$$

in which the kinetic overpotential:

$$\eta = \phi_s - \phi_e - U_i(c_{s,i}) - iR_f, \quad (S8)$$

and exchange current density:

$$i_0 = k(T) c_{s,i}^{\alpha_c} c_e^{\alpha_a} (c_{s,max} - c_{s,i})^{\alpha_a}. \quad (S9)$$

The reaction current density on particle surface and volumetric current density in the electrodes are related:

$$j = ai, a = 3 * (1 - \varepsilon) / r_i. \quad (S10)$$

The heat generation power:

$$q = j(\phi_s - \phi_e - U) + \sigma_s^{eff} \nabla \phi_s \cdot \nabla \phi_s + \kappa^{eff} \nabla \phi_e \cdot \nabla \phi_e + \kappa_D^{eff} \nabla \ln c_e \cdot \nabla \phi_e + j(T \frac{dU}{dT}). \quad (S11)$$

The heat generation comes from four terms: $j(\phi_s - \phi_e - U)$ represents kinetic heat, $\sigma_s^{eff} \nabla \phi_s \cdot \nabla \phi_s$, $\kappa^{eff} \nabla \phi_e \cdot \nabla \phi_e$ and $\kappa_D^{eff} \nabla \ln c_e \cdot \nabla \phi_e$ are joule heat from electronic resistance, ionic resistance and concentration overpotential respectively, and $j(T \frac{dU}{dT})$ is the reversible heat.

To verify our ECT model, we simulated the anode potential during XFC and observed a good agreement with the experiment results from a three-electrode cell (see Supplementary Figures 1a-c). In this verification study, we assembled three-electrode cells using materials as described in our prior work⁹.

For the ECT simulation of a representative commercial battery, *e.g.*, commercial LCO/C cells, the thermal model uses the heat generation rate from the electrochemical simulation as input, and the evolution of battery temperature during charging in different thermal conditions is calculated. We performed 3D transient heat transfer simulations for the battery with or without BTMS. The temperature distribution inside the battery (T_B) is governed by the Fourier's law:

$$\rho C_p \frac{\partial T_B}{\partial t} = \frac{\partial}{\partial x} (k_x \frac{\partial T_B}{\partial x}) + \frac{\partial}{\partial y} (k_y \frac{\partial T_B}{\partial y}) + \frac{\partial}{\partial z} (k_z \frac{\partial T_B}{\partial z}) + q, \quad (S12)$$

where ρ , C_p , k , and q are the density, heat capacity, thermal conductivity, and volumetric heat generation rate of the battery, respectively. This equation was solved with the coupled heat generation rate and the convective heat transfer boundary condition on the surface. Details on the calculation of heat generation rate can be found in prior works^{10,11}. For the simulation of cell packs including BTMS, the BTMS was an additional thermal mass resulting in heat transfer with the battery (*e.g.*, heat leakage from the battery to the heat sink). According to Yang *et. al.*¹⁰, the gravimetric cell-to-pack ratio is 55-65%, which means 35-45% of the pack weight is taken by management system, metals, cabling and others. We estimate the heat capacity of management system as the average of aluminum and coolants (water/glycol)¹².

Gen2 electrolyte (1.2 M LiPF₆ in EC:EMC 3:7) was used in all the studies, which is the baseline electrolyte for XFC suggested by the US DOE. We used the transport properties of

electrolyte, graphite and LiCoO₂, e.g., diffusion coefficient and conductivity, provided by the COMSOL material library. The open circuit voltage (OCV) of graphite and LiCoO₂ were also from the COMSOL material library.

A representative 3D temperature distribution at the end of 6C charge when $h = 10 \text{ W/m}^2\text{K}$

Electrode

Parameter	Anode (Graphite)	Cathode (LiCoO ₂)
Thickness (μm)	55.4	52.2
Initial Porosity	0.382	0.356
Loading (mAh/cm^2)	2.34	2.11
Particle radius (μm)	5	5 ¹³
Specific surface area (m^{-1})	3.267×10^5	2.97×10^5

Usable capacity range by lithium intercalation fraction	0.12~0.95	0.453~0.994
Bruggeman exponent, p	2.55	2.2
Reference exchange current density (A/m^2)	2.1^{13}	2.1
Activation energy of exchange current density (kJ/mol)	68^{13}	69^{14}
Solid-state diffusivity, $D_s(cm^2/s)$	COMSOL Materials Library	COMSOL Materials Library
Activation energy of solid-state diffusivity (kJ/mol)		25^{15}

Separator

Electrolyte concentration (mol/L)	1.2
Thickness (μm)	25
Porosity	0.41
Bruggeman exponent, p	2

Cell

Cathode Material	LiCoO ₂
Specific heat of cell, $c_p (J/(kg * K))$	1100^{16}
Mass of the battery (kg)	0.211
Thickness (mm)	13.5
Length (mm)	90
Width (mm)	61
In-plane thermal conductivity ($W/(m*K)$)	25^{11}
Cross-plane thermal conductivity ($W/(m*K)$)	0.8^{11}

Updated Supplementary Figure 1d and e using the 3D simulation results:

Supplementary Figure 1. Evaluation of thermal strategy using ECT model. Verification of our ECT model by comparing to the anode potential measured in a three-electrode cell during charging at **a)** C/3, **b)** 1C, and **c)** 6C. Evolution of battery temperature and anode potential during XFC of LCO/C cells using **d)** the system-level thermal strategy and **e)** the local thermal control of the cell. We use the global thermal conductance per unit area (h_{gl}) ranging from 10 W/m²-K to 1000 W/m²-K to mimic the different status of coolant flow. Regardless of the flow status, the heat leakage from the cell to the BTMS reduces the battery temperature rise during XFC and results in the negative anode potential. In contrast, the minimum anode potential can maintain positive by local thermal control of the cell (*i.e.*, minimized heat leakage from the battery to the BTMS) with $h \sim 10$ W/m²-K.

Updated Supplementary Figure 3 using the 3D simulation results:

Supplementary Figure 3. Maximum battery temperature and minimum anode potential during XFC using our validated ECT model as a function of thermal conductance per unit area (h) for the cell. The minimum anode potential was positive during XFC when the cell is thermally insulated with $h \sim 10 \text{ W/m}^2\text{-K}$.

In transient heat transfer, thermal boundary conditions are not that important, as the heat loss equals to multiplication of heat transfer coefficient, surface area, temperature difference, and the time. If the time is short as in this application, the total heat loss is small regardless of heat transfer coefficient (i.e. fast transient heat transfer).

If preheating time is so short, say less than 1 minute, why does the battery need to be thermally insulated (Line 62). I cannot understand this from the principles of heat transfer. I suggest that the authors bring in additional heat transfer experts into the team to fully analyze thermal energy balance and clear fundamental mistakes before this work can be published.

Response:

This is a very good point raised by the reviewer. From our 3D transient heat transfer analysis, the temperature profile is similar from SOC = 0 to 10% during 6C charging (i.e., ~1 min) regardless of the effective heat transfer coefficient (Supplementary Figure 1 d and e) as astutely observed by the reviewer. However, the role of thermal boundary conditions becomes significant at higher SOC's (i.e., >1 min) and causes the different temperature profile. Since **the charge time associated with 6C CCCV charging to 80% SOC is 10 to 15 min**, the thermal boundary conditions have a huge impact on the battery temperature during fast charging, as predicted and shown in Supplementary Figure 1 d and e.

Supplementary Figure 1. Evaluation of thermal strategy using ECT model. Verification of our ECT model by comparing to the anode potential measured in a three-electrode cell during charging at **a)** C/3, **b)** 1C, and **c)** 6C. Evolution of battery temperature and anode potential during XFC of LCO/C cells using **d)** the system-level thermal strategy and **e)** the local thermal control of the cell. We use the global thermal conductance per unit area (h_{gl}) ranging from $10 \text{ W/m}^2\text{-K}$ to $1000 \text{ W/m}^2\text{-K}$ to mimic the different status of coolant flow. Regardless of the flow status, the heat leakage from the cell to the BTMS reduces the battery temperature rise during XFC and results in the negative anode potential. In contrast, the minimum anode potential can maintain positive by local thermal control of the cell (*i.e.*, minimized heat leakage from the battery to the BTMS) with $h \sim 10 \text{ W/m}^2\text{-K}$.

This also answers the reviewer's question regarding the need of thermal insulation in Ref. 21 (Line 62 in the original version; Line 65 and 66 in the revised manuscript). Since the time scale for fast charging is 10 to 15 min, the heat loss needs to be reduced by thermal insulation for maintaining the high battery temperature during the whole 10 to 15 min charging process. We performed 3D heat transfer simulation to verify that the high initial battery temperature (*e.g.*, $50 \text{ }^\circ\text{C}$) is not sufficient for maintaining the high battery temperature during fast charging if the battery is not appropriately thermally insulated (see Figure R1).

Figure R1. a, Evolution of battery temperature during 6C charging with an initial T_B of 50 °C and different heat transfer coefficient. The battery temperature can decrease during the 10-15 min fast charging process if the battery is not appropriately thermally insulated (e.g., $h = 100$ W/m²K), resulting negative anode potential during charging as shown in panel (b).

Reviewer #2 (Remarks to the Author):

The active thermal switching structure can retain the heat during XFC with the switch OFF to boost the kinetics while dissipating the heat after XFC with the switch ON to reduce side reactions. There are still some issues that need to be addressed in the application of battery thermal management.

1. How is the active switch increases the battery's temperature during charging? In my opinion, the active switch is used to control the heat dissipation to adjust the battery temperature passively.

Response:

The reviewer is correct that the battery temperature is modulated as the heat dissipation is controlled via a switch. The battery heat is either dissipated to the heat sink or stored in the battery itself. The latter term is simply a product of mass, heat capacity, and temperature. Based on energy balance, the heat stored in the battery increases as the heat dissipation is minimized with our switch, leading to an increase of battery temperature.

Revision:

We clarified this relationship in the revised manuscript as (page 2 and 3):

“Treating the battery as a lumped thermal system (see Supplementary Note 1 for details on the validity of lumped model), the transient battery temperature can be written as $T_B(t) = (Q - mC_p \frac{\partial T_B}{\partial t})/hA + T_c$, where Q , A , T_B , m , C_p are the transient heat generation, surface area, temperature, mass, heat capacity of the battery, respectively. T_c is the coolant temperature and h denotes the tunable thermal conductance per unit area between the battery and coolant, as illustrated in Fig. 1a. Thus, elevating the battery temperature relies on the increased Q and T_c and/or the reduced hA (see Supplementary Note 1 for the 3D transient electrochemical-thermal model).”

2. If the battery is in a low temperature for a long time, can the active switch increase the battery's temperature?

Response:

The thermal equilibrium effect has been considered in our study. Our charge protocol includes a sufficient long rest after discharge so that the battery temperature is almost the same as the ambient temperature before next XFC cycle. Thermal equilibrium is restored for each cycle and increasing the rest time further does not change the thermal condition and battery temperature. The relevant text can be found in the Methods:

“The rest time after discharge is 30 min for the case of insulation and 15 min for other cases, which ensures thermal equilibrium at the end of each XFC cycle, e.g., <1 °C temperature difference compared with the ambient temperature.”

3. From a practical standpoint, can this active switch structure used in a battery pack? The extra gap volume affects the heat transfer between batteries and it is not permitted in a battery pack.

Response:

We proposed a pack-level design in Supplementary Figure 19 as mentioned below by the reviewer. Although pack-level manufacturing and tests are beyond the experimental capability of our lab,

we would like to point out that our nonintrusive approach does not rely on the modification of battery cells and is advantageous for the application at different scales.

As for the gap volume, our switch between the battery and heat sink does not involve any modification of the contact between batteries, and thus does not affect the heat transfer between batteries. As shown in the battery pack solid model (Fig. 19 in the Supplementary Information) there is no gap between various batteries. The switch only operates between the battery pack and the heat sink covering the pack. Therefore, there is no impact to the battery pack. In addition, the gap between the battery and heat sink is closed in the ON state for efficient heat dissipation during discharging with an effective heat transfer coefficient comparable to that for the cooling case with no gap, as discussed in the manuscript (page 7).

4. Fig.S19 shows the pack-level design for active thermal switching. However, the heat dissipation surface of the pack-level is the side-surface of the batteries, while that of single battery is the front-surface of the battery. Does the active thermal switch still have good thermal control performance when the side-surface of the battery is in contact with the cold plate?

Response & Revision:

The confusion partly comes from the not-to-scale schematic in Supplementary Figure 19. We clarified this in the legend: “Parts in the schematic are not to scale and can be adapted to batteries of different sizes”. We further highlighted this in the revised manuscript: “We anticipate that our switch can be adapted to batteries and packs of different sizes due to the nonintrusive nature”.

Supplementary Figure 19. Pack-level design for active thermal switching. At the pack or module level, the gap between the battery and cold plate can be controlled by moving the cold plate via heating the SMA wires. To switch from ON to OFF, the self-heated SMA wires contract and open the gap between the cold plate and battery. The application of our approach to the pack level requires minor modifications which benefits from the nonintrusive nature. Parts in the schematic are not to scale and can be adapted to batteries of different sizes.

As for the surface used in battery thermal management, we note that either surface can be used for cooling/heating in existing commercial battery packs, depending on the geometry and other factors. For example, BYD Auto's Blade battery system puts the cold plate cooling on top of the battery pack which are in contact with the side-surface of the cell (source: <https://www.batterydesign.net/byd-blade/>). To answer the reviewer's question, we performed both simple thermal resistance analysis and 3D ECT simulations of a representative commercial pack with prismatic cells which uses the side surface for heating/cooling. The simulation results support that our thermal switching strategy works for this type of cell and thermal management and can effectively cool down the battery after fast charging (see Figure R2). This is anticipated as the thermal management associated with the exact battery surface is appropriately designed for commercial applications. Our approach simply provides the low h for XFC. Thus, the use of different surfaces does not affect the effectiveness of our method.

Figure R2. Schematics of (a) the commercial battery pack using the side surface of batteries for heating/cooling and (b) the geometry of a representative prismatic cell (36.7 cm \times 8.65 cm \times 2.8 cm) used in our simulation. Details on the other parameters and simulation can be found in Supplementary Note 1. For this cell, the thermal resistance associated with the front-surface heat transfer is 0.55 K/W, which is much higher than the thermal resistance related to the side surface (*i.e.*, 0.17 K/W). This is a result of the cell geometry and the high thermal conductivity anisotropy (*i.e.*, the in-plane thermal conductivity is much higher than the cross-plane thermal

conductivity). Thus, the side surface is used for thermal management in this type of cell. As a verification, we performed 3D heat transfer simulation to evaluate the performance of cooling using the side surface. The heat transfer coefficient corresponding to the side surface is denoted as h_s . The predicted evolution of (c) average battery temperature and (d) temperature difference proves the effectiveness of side-surface cooling for the cell with a certain geometry. The battery was firstly 6C fast charged to 80% SOC under insulation condition which brought it to a high temperature. Then the switch was turned ON to dissipate the heat. With h_s in the range of forced convection, the battery could be cooled down to <40 °C in a short time, *e.g.*, the battery drops below 36 °C in 10 mins with $h_s = 500W/(m^2K)$.

REVIEWER COMMENTS

Reviewer #1 (Remarks to the Author):

The authors have satisfactorily answered my questions. In particular, I find that Figure R1(a) is very useful and insightful for readers. It is interesting to see from Figure Ra(a) that a heat transfer coefficient between 10 and 100 W/m²K would nicely maintain the battery temperature. This can be easily realized by variable air flow (i.e. changing air velocity during air cooling). If the authors agree to add Figure R1 and related discussions in the rebuttal letter into the final manuscript, and furthermore point out that variable air flow can be an alternative active switch as proposed in this manuscript, I'd be happy to recommend publication of this work.

A quick comment about the authors' reply to Question #1 of Reviewer #2: if the battery is in a low temperature, say -20 deg C, for a long time, the proposed active switch in thermal boundary conditions obviously does not work for XFC, in my humble opinion. This is an obvious conclusion, and it is not a problem to admit limitation of a new technique.

We thank the Reviewers for their time and appreciate their valuable suggestions to improve the manuscript. Here we provide a detailed point-by-point response to the Reviewers' comments, and we have edited the manuscript and Supplementary Information accordingly.

Color codes used in this response letter:

Black Italic: original review comments;

Blue: our responses;

Red: revisions made in the manuscript.

Reviewer #1 (Remarks to the Author):

The authors have satisfactorily answered my questions. In particular, I find that Figure R1(a) is very useful and insightful for readers. It is interesting to see from Figure Ra(a) that a heat transfer coefficient between 10 and 100 W/m²K would nicely maintain the battery temperature. This can be easily realized by variable air flow (i.e. changing air velocity during air cooling). If the authors agree to add Figure R1 and related discussions in the rebuttal letter into the final manuscript, and furthermore point out that variable air flow can be an alternative active switch as proposed in this manuscript, I'd be happy to recommend publication of this work.

Response:

We thank the reviewer for the positive evaluation. We agree that Figure R1 and related discussions are useful for readers and should be added to the manuscript. And the use of air flow control for thermal switching in air cooling systems is a good point to be added in the revised version. Also, we point out that liquid cooling is used in most existing electric vehicles for many reasons, e.g., pack-level temperature uniformity. Our thermal switch between the heat sink and battery is designed for this mainstream scenario.

Revision:

We added Figure R1 and related discussion (Page 6):

“Since the time scale for XFC is 10 to 15 min, a high initial temperature (e.g., 50 °C) is not sufficient for maintaining the high battery temperature during the whole charging process if the battery is not appropriately thermally insulated (Fig. 1b-c).”

Added Figure R1 into Fig. 1b-c and the related panel number was updated.

We discussed the use of thermal switching strategy in different BTMSs including air cooling system, and pointed out that thermal switching can be easily realized by controlling the air flow velocity in air cooling systems (Page 17):

“In addition, the thermal switching strategy is universal to various BTMSs. Here, our SMA-based thermal switch is developed for decoupling the thermal mass of the battery and the cold plate (or heat sink) in the mainstream liquid cooling system. Note that liquid cooling is used in most EVs for many reasons such as the pack-level temperature uniformity. For other BTMSs, e.g., air cooling systems, thermal switching can be potentially realized by controlling the air flow velocity.”

A quick comment about the authors' reply to Question #1 of Reviewer #2: if the battery is in a low temperature, say -20 deg C, for a long time, the proposed active switch in thermal boundary

conditions obviously does not work for XFC, in my humble opinion. This is an obvious conclusion, and it is not a problem to admit limitation of a new technique.

Response:

We agree with the reviewer that our thermal switching strategy alone cannot achieve XFC at such low ambient temperature. We discussed this in the revised manuscript.

Revision:

We added this discussion in the revised manuscript (Page 15):

“At extremely low T_a (e.g., -20 °C), fast charging is particularly difficult given the challenge of XFC even at room temperature. Our thermal switching strategy alone cannot achieve XFC at such low T_a . We anticipate that realizing such a goal requires extensive multilevel research and development including thermal management.”

Reviewer #1 confidential remark to the editor based on comment no. 3 and 4 from Reviewer #2 (Remarks to the Author):

Comment 3 from reviewer 2: From a practical standpoint, can this active switch structure used in a battery pack? The extra gap volume affects the heat transfer between batteries and it is not permitted in a battery pack.

Response:

We have modified the discussion in the manuscript to clarify things further.

Revision:

We clarified the impact of switch on heat transfer between batteries, and pointed out that the relative volume, mass, and material cost of the pack-level switch is small (Page 16):

“Like the device-level switch, this proposed pack-level thermal switch only operates between the battery pack and the cold plate and thus does not affect the heat transfer between batteries. The relative volume, mass, and material cost of the pack-level switch is estimated to be comparable to that of the device-level thermal switch, which shows the promise of practical applications.”

We also moved the pack-level design in Supplementary Figure 19 in the previous version to the revised manuscript, as shown in Figure 5 (Page 16-17):

“At the pack level, the contact and separation between the cells and cold plate can be controlled in a similar manner, *e.g.*, moving the cold plate via the contraction of SMA wires (see Figure 5 for a potential pack-level design).”

Fig. 5 | A potential pack-level design for active thermal switching. At the pack or module level, the gap between the battery and cold plate can be controlled by moving the cold plate via heating the SMA wires. To switch from ON to OFF, the self-heated SMA wires contract and open the gap between the cold plate and battery. The

application of our approach to the pack level requires minor modifications which benefits from the nonintrusive nature. Parts in the schematic are not to scale and can be adapted to batteries of different sizes.

Comment # 4 from reviewer 2: Fig.S19 shows the pack-level design for active thermal switching. However, the heat dissipation surface of the pack-level is the side-surface of the batteries, while that of single battery is the front-surface of the battery. Does the active thermal switch still have good thermal control performance when the side-surface of the battery is in contact with the cold plate?

Response:

The confusion partly comes from the not-to-scale schematic in Supplementary Figure 19 in the previous version. Our thermal switch can achieve the contact or separation regardless of the surface used for heat transfer.

As for the surface used in battery thermal management, we note that either surface can be used for cooling/heating in existing commercial battery packs, largely depending on the geometry. For example, BYD Auto's Blade battery system puts the cold plate on top of the battery pack which is in contact with the side surface of the cell (source: <https://www.batterydesign.net/byd-blade/>). To answer the reviewer's question, we performed both simple thermal resistance analysis and 3D ECT simulations of a representative commercial pack with prismatic cells which uses the side surface for heating/cooling. The simulation results support that our thermal switching strategy works for this type of cell and thermal management and can effectively cool down the battery after fast charging (see **Supplementary Figure 19**). This is anticipated as the thermal management associated with the exact battery surface is appropriately designed for commercial applications at normal rates. Our approach simply provides the low h for XFC. Thus, the use of different surfaces does not affect the effectiveness of our method.

Revision:

We discussed the use of front or side surfaces in batteries of different sizes (Page 16), and added the ECT simulation results to the revised SI as Supplement Figure 19:

“Depending on the battery geometry, the front or side surface is selected as the contact surface for cooling/heating in commercial battery packs. Our thermal switch can achieve the contact or separation regardless of the surface used for heat transfer. The effectiveness of our strategy for thermal management using the side surface of batteries is verified by ECT simulations, as shown in Supplementary Figure 19. ”

Updated Supplementary Figure 19 in the revised SI:

Supplementary Figure 19. Active thermal switching at the side surface. Schematics of (a) the commercial battery pack using the side surface of batteries for heating/cooling and (b) the geometry of a representative prismatic cell ($36.7 \text{ cm} \times 8.65 \text{ cm} \times 2.8 \text{ cm}$) used in our simulation. Details on the other parameters and simulation can be found in Supplementary Note 1. For this cell, the thermal resistance associated with the front-surface heat transfer is 0.55 K/W , which is much higher than the thermal resistance related to the side surface (*i.e.*, 0.17 K/W). This is a result of the cell geometry and the high thermal conductivity anisotropy (*i.e.*, the in-plane thermal conductivity is much higher than the cross-plane thermal conductivity). Thus, the side surface is used for thermal management in this type of cell. As a verification, we performed 3D heat transfer simulation to evaluate the performance of cooling using the side surface. The heat transfer coefficient corresponding to the side surface is denoted as h_s . The predicted evolution of (c) average battery temperature and (d) temperature difference proves the effectiveness of side-surface cooling for the cell with a certain geometry. The battery was firstly 6C fast charged to 80% SOC with switch OFF which brought it to a high temperature. Then the switch was turned ON to dissipate the heat. With h_s in the range of forced convection, the battery could be cooled down to $<40 \text{ }^\circ\text{C}$ in a short time, *e.g.*, the battery drops below $36 \text{ }^\circ\text{C}$ in 10 mins with $h_s = 500 \text{ W}/(\text{m}^2 \text{K})$.

REVIEWERS' COMMENTS

Reviewer #1 (Remarks to the Author):

recommend publication